# The Feasibility of the Functional Listening Index—Paediatric (FLI-P^®^) for Young Children with Hearing Loss

**DOI:** 10.3390/jcm11102764

**Published:** 2022-05-13

**Authors:** Aleisha Davis, Elisabeth Harrison, Robert Cowan

**Affiliations:** 1Department of Linguistics, Macquarie University, Sydney 2109, Australia; elisabeth.harrison@mq.edu.au (E.H.); r.cowan@unimelb.edu.au (R.C.); 2The Shepherd Centre, 146 Burren Street, Sydney 2042, Australia; 3Department of Audiology and Speech Pathology, The University of Melbourne, Melbourne 3010, Australia

**Keywords:** FLI-P, functional listening, early intervention, hearing loss, pediatric, outcomes, tracking progress

## Abstract

(1) Background: There is clear evidence supporting the need for individualized early intervention in children with hearing loss. However, relying on hearing thresholds and speech and language test results to guide intervention alone is problematic, particularly in infants and young children. This study aimed to establish the feasibility of a tool to monitor the development of functional listening skills to inform early and ongoing decisions by parents and professionals. (2) Methods: The FLI-P^®^ is a 64-item checklist completed by parents and/or a child’s team. The listening development of 543 children with hearing loss enrolled in an early intervention and cochlear implant program was tracked with the FLI-P over a 6-year period. The scores for individual children were grouped according to hearing loss, device, additional needs, and age at device fitting. (3) Results: Results indicate that the FLI-P is a feasible and viable clinical measure that can be used to identify and track a child’s developing listening skills. Its use across a wide range of children supports its broad application. Children’s individual scores and aggregated group data were consistent with indicated expected differences and variations. Children’s individual scores and aggregated group data indicated expected differences and variations. (4) Conclusions: Information provided by children’s listening scores on the FLI-P can guide and support discussions and intervention decisions and bridge the gap between information from audiological assessments and language measures.

## 1. Introduction

The language that children develop from infancy results from one of the most remarkable human developmental processes. As a specialized and complex skill, language develops spontaneously without conscious effort or formal instruction, is deployed innately without awareness of its underlying logic, is qualitatively the same in all individuals, and is distinct from more general abilities to process information or behave intelligently [1]. However, for children with hearing loss, language development does not happen without specific and targeted intervention. The detrimental impact of hearing loss on the development of language and communication skills has been well documented [2,3,4]. Despite advances in the early detection of hearing loss, a hearing-impaired child’s lexical acquisition and language development is still impacted [5]. Given the impact that communication abilities have on the quality of life, educational attainment, future earnings, life opportunities, the use of health care systems, and life expectancy [6,7], improving early intervention practices for children with hearing loss is of considerable importance.

Determining a child’s access to sound through the measurement of their threshold responses to frequencies across the speech spectrum does not provide information on their use of sound (identification), nor their integration of sound into meaning (comprehension) [8]. Access and detection do not automatically imply an ability to cognitively identify a sound, understand how it is being used, and subsequently be able to produce it correctly in context nor respond appropriately and meaningfully to the information being conveyed acoustically. Whilst individual measures of detection and discrimination establish the level of access a child has to auditory information such as in behavioral and objective electrophysiological assessments, they do not necessarily ascertain whether this information results in the child developing a relationship between individual sounds and their meanings (i.e., functional listening). As such, understanding how a child integrates and uses the sounds to which they have access, is as important as detecting or hearing the sounds themselves in ensuring their listening skill acquisition is on track to appropriately support language and communication development.

Despite considerable investment in the research, design, and development of hearing devices and coding strategies [9,10], it is also difficult to accurately evaluate the effect of device fitting and audibility levels over time in infants and young children. Moreover, as language in typical hearing children develops over time as the child ages, having a single measure that is appropriate across this wide range of age and rapid development is a key need, which has not yet been filled. Even with a wide range of auditory measures available, in practice there are limitations to their use. These include the lack of versatility across age ranges, limited incorporation of real-world skills, minimal detail of how sound is used at a cognitive level, and the lack of ability to visually track progress and provide guidance for next steps in intervention.

A child’s progress in early intervention services is often routinely measured by standardized language assessments. Between birth and 3 years of age these assessments are criterion-referenced parental checklists of receptive and expressive language. In older children, language assessment results become more reliable with the use of standardized and norm-referenced assessments completed by the child. Current clinical practices rely heavily on these results for information to guide intervention decisions with families. These decisions may be regarding amplification, or to address family context or educational factors that may be impacting outcomes. Often, changes or decisions about intervention are prompted when standardized assessment results show poorer than expected progress in comparison to typical hearing peers. When a child with hearing loss is attempting to ‘close the gap’ to catch up to the progress of typical hearing peers, the timeframe in making decisions is critical and 6 month or annual assessments become problematic. The Early Hearing Detection and Intervention framework [11], highlights the impact of timing in the communication development of children with hearing loss and highlights the need for decisions at the earliest opportunity. For parents and professionals to feel confident in making timely decisions they need meaningful and interpretable information that is specific to the child.

How a child with hearing loss detects, uses, and processes linguistic input in their everyday settings, that is, their ‘functional listening skills’, is critical to understanding how well they are able to develop oral language. As such, the Functional Listening Index (FLI-P^®^) was developed to track the acquisition of a child’s listening skills over time and provide a trajectory of developing listening competency. This information could be used by parents and caregivers to inform and guide early decisions, enabling and empowering choices regarding their child’s intervention. Similarly, such information could be used by professionals to monitor progress and optimize intervention through targeted listening, learning, and language experiences in a child’s early and critical development years. Tracking functional listening acquisition through such a tool may have the potential to improve a child’s language and communication outcomes through informed, timely decisions, and, individually, appropriately targeted intervention.

The aim of this clinical retrospective study was to determine the feasibility and viability of the FLI-P as a clinical measure for all children with hearing loss enrolled in an early intervention and cochlear implant program, through three research objectives:Can the FLI-P be used successfully with all children with hearing loss attending an early intervention/clinical service?Do children’s individual FLI-P trajectories change over time as would be expected?Do the data for known groups show the expected differences (typical hearing versus hearing loss, bilateral versus unilateral hearing loss, presence of an additional disability to hearing, ANSD, age at diagnosis, type of device, level of loss, age of implant)?

## 2. Materials and Methods

### 2.1. Participants

Two groups of children participated in this feasibility study. The first group were children with hearing loss (HL) aged between 0 and 72 months, enrolled in an integrated early intervention and cochlear implant program in Australia (EI group) (*n* = 543). This particular early intervention program provided family-centered early intervention services across three states of Australia for families choosing a listening and spoken language communication approach. Families attended individual and/or group sessions at varying degrees of frequency dependent upon individual need. Weekly intervention frequency was typical for children under 3 years of age with bilateral moderate hearing loss or greater. For children 4 to 5 years of age, group sessions often took the place of individual therapy sessions if listening and language development was progressing well, with intervention focusing on social skills in noisy, real-life environments, and readiness for school. 

The second group of participant children all had typical hearing (TH) (whereby typical hearing refers to hearing that was screened and hearing thresholds were within 20 dB or less) (*n* = 32). This group was a convenience sample of children who were either attending the preschool program for typical hearing children at two of the early intervention centers (*n* = 20), or children of the research and clinical team members (*n* = 12). The TH group served as an initial group to provide benchmark functional listening skills for children with typical hearing. 

### 2.2. Early Intervention (EI) Group

Demographic and audiological information was collected during routine clinical services. Demographic information included date of birth, gender, and presence of additional needs. Audiological information collected included Universal Newborn Hearing Screening (UNHS) result, date of diagnosis, date of device fitting, level of hearing loss (left and right at 500 Hz, 1 kHz, 2 kHz, 4 kHz), type of hearing loss (left and right), device type (left and right), FLI-P scores, and date of cochlear implant/s surgery. Audiological and demographic characteristics of the EI group are provided in Table 1, Table 2, Table 3, Table 4, Table 5, Table 6 and Table 7. The only children excluded from the study were newly enrolled to the service who did not yet have the FLI-P administered as part of their routine clinical protocols. Throughout the duration of the study, there were no children reported by the clinical team who were unable to have their listening skills measured using the FLI-P.

The dataset included children from a variety of language backgrounds, classified as either monolingual children who spoke English only, bilingual children who spoke English as a primary language, bilingual children who spoke English as an additional language, or monolingual children who spoke only a language other than English. Where necessary, the FLI-P was administered to families in non-English languages through an interpreter. All children enrolled in the program had a permanent sensorineural and/or conductive hearing loss (hearing thresholds greater than 25 dBHL inclusive [12]), in one or both ears. Levels of hearing loss ranged from mild through to profound, and included both unilateral and bilateral hearing losses. Children in the program were fitted with a range of device types and configurations including monaural, binaural, and bone anchored hearing aids; unilateral and bilateral cochlear implants; and bone anchored implantable devices. One in five children (20%) enrolled in the program had a diagnosed disability in addition to hearing loss. 

### 2.3. Typical Hearing (TH) Group

Parents of children in the TH group (*n* = 32) reported no concerns for their child’s speech and language development or additional needs that impacted on learning. All children completed the Clinical Evaluation of Language Fundamentals—Fourth Edition, Screening Test Australian & New Zealand Language Adapted Edition [13] to ensure language was at age-appropriate levels. There were five children who failed the screening test and were excluded from the study, resulting in 27 children in the TH group.

### 2.4. Procedures: EI Group

The FLI-P was administered to all children in the EI group. Administration of the FLI-P was completed by their case manager in collaboration with the child’s family. The case manager was either a pediatric audiologist, or listening and spoken language therapist/specialist with either speech pathology or teacher of the deaf qualifications. The FLI-P was re-administered to each child every three or four months and data entered in the clinical database. In some cases, the FLI-P was used more regularly, for example when a team member was concerned about progress or development, for specific populations, or in situations of rapid increase, or decline, in listening skill.

### 2.5. Procedures: TH Group

The FLI-P was administered by a clinician experienced with the tool, either at one time or on multiple occasions every three or four months.

### 2.6. Definitions

The presence of additional needs was recorded if a formal written diagnosis had been received, and if it was considered by the case manager to impact on learning or language development. Age of implant was defined as the date of the child’s (first) surgery. Hearing loss level was categorized according to a child’s hearing in their better ear and defined as mild = 26–40 dB, moderate = 41–55 dB, moderately severe = 56–70 dB, severe = 71–80 dB, and profound ≥ 91 dB [14,15]. Auditory Neuropathy Spectrum Disorder (ANSD) was defined as a type of hearing loss rather than an additional need, and not included in the level of hearing loss classifications.

### 2.7. Data Collection, Extraction and Validation

FLI-P scores were collected for the EI group between August 2012 and February 2018, and extracted from the clinical database in March 2018. The data were verified for validation and accuracy and incorrect scores removed. FLI-P scores for the TH group were collected between January 2014 and February 2018. Blank fields in the clinical database were reported as unknown and categorized accordingly. 

### 2.8. Data Analysis

A total of 2869 FLI-P results were collected for the 543 children in the EI group (mean = 5.3, SD = 4.3) and 51 FLI-P results for the 27 children in the TH group (mean = 5.2, SD = 1.7). Results were exported from the database for analysis by demographic and audiological factors. Group data were aggregated and analyzed for differences between typical hearing and hearing loss, presence of a disability in addition to hearing loss, bilateral and unilateral hearing loss, presence of ANSD, age of diagnosis, type of device, level of hearing loss, and age of implant. FLI-P assessments which were incomplete were removed from the dataset (*n* = 27).

## 3. Results

### 3.1. Use of the FLI-P in a Clinical/Educational Setting

The listening skills of all children with hearing loss enrolled in the intervention program (EI group) during the period of the study were able to be measured using the FLI-P. All FLI-P results are displayed in Figure 1 by each child’s age in months. Results indicate an increase in FLI-P scores with age, with wide variability as would be expected for the heterogeneity of the population of children with hearing loss in the program.

### 3.2. Changes to Children’s Individual Scores over Time

Each child’s FLI-P results were collected over the time period of the study and used to build individual listening trajectories. The trajectories of children with a bilateral hearing loss and 5 or more recorded FLI-P scores (*n* = 257) were graphed according to the level of hearing loss (mild, mild–moderate, and mild–profound; moderate and moderate–severe; severe and severe–profound; and profound). Children with only high frequency hearing loss were included in the mild and in the mild to profound group for the purposes of the analysis. Children with a developmental need in addition to hearing loss, ANSD in one or both ears and asymmetrical hearing losses were excluded from this analysis for comparative purposes. 

Individual trajectories for each group are displayed against all FLI-P results for the EI group in Figure 2. Each of the colored lines represents an individual child’s trajectory to highlight progress and different listening development trajectories. Growth in listening skills over time with age was as expected. Steep inclines in trajectories indicated rapid acquisition of skills. Variability in trajectories increases with the level of hearing loss with the most varied FLI-P trajectories for children with profound hearing loss (Figure 2d).

#### 3.2.1. Age of Implant

Children’s individual listening development by FLI-P score over time was tracked and compared by the age of first implant (<6 months, 6–11 months, and 12–23 months). FLI-P scores for children with a disability in addition to hearing loss or ANSD in one or both ears were excluded given their impact on outcomes (Figure 3). Trajectories for children who received a cochlear implant <6 months of age show the smallest amount of variation with listening skills at the top levels of the EI group, with increased variability in FLI-P scores with age for children who received an implant at older ages, with each colored line representing an individual child’s trajectory. Different rates of acquisition at stages are indicated, ranging from slow development of listening skills with the quickest acquisition of skills by children receiving the earliest implants.

#### 3.2.2. Developmental Need in Addition to Hearing Loss

The development of listening skills of all children with an additional disability in the program was measured routinely using the FLI-P. The individual trajectories of children with additional needs and a cochlear implant are provided in Figure 4 (*n* = 14). Trajectories indicate sensitivity of the FLI-P to individual progress over time and range of rates of skill development. Some rapid gains in skills can be observed at early ages after implantation and slower periods of acquisition indicated by flat stages in individual trajectories.

#### 3.2.3. Auditory Neuropathy Spectrum Disorder

The trajectories of functional listening development in individual children with bilateral ANSD were graphed by FLI-P score over time (*n* = 10) compared to children with unilateral ANSD (no hearing loss in the other ear) (*n* = 10) (Figure 5). Results for both bilateral and unilateral ANSD indicated a spread of listening skills by age, as would be expected with different degrees of neuropathy and the range of ANSD in clinical presentations. Despite typical hearing levels in one ear for children with unilateral ANSD, FLI-P scores show varied rates of listening development. 

### 3.3. Differences in Group Scores

#### 3.3.1. TH Group/EI Group

Group data of children’s FLI-P scores were aggregated to determine if the expected differences were observed between the FLI-P scores for children in the EI group to the FLI-P scores for children in the TH group. There were 46 FLI-P results collected for 27 children in the TH group, and the age at data collection ranged from 2 to 63 months (average age 32 months). A single FLI-P result was collected for 20 children, 2 data points for four children and 3, 5, and 10 data points for the other three children in the group. The lowest FLI-P score for the TH group was 2 items at 1 month of age, and the highest score was 49 items (of the 60 in total) at 63 months of age. The FLI-P results for the TH group are graphed against age in months in Figure 6. A line of best fit indicates significant correlations over time ($R^{2}$ = 0.92).

A comparison of FLI-P scores for children in the EI and TH groups indicate expected differences (larger variation and less developed listening skills across age ranges excluding children with an additional disability for the purposes of comparison). Results demonstrate that a number of children with hearing loss achieved similar FLI-P scores to children in the TH group. Lines of best fit indicate significant linear relationships for both the TH group ($R^{2}$ = 0.80) and the EI group ($R^{2}$ = 0.70), despite large numbers and variability in the EI group.

#### 3.3.2. Bilateral and Unilateral Hearing Loss

The FLI-P results for children with bilateral hearing loss (*n* = 385, 2130 FLI-P scores) were compared to FLI-P results for children with unilateral hearing loss (*n* = 140, 696 FLI-P scores) (Figure 7). Results indicate similar FLI-P scores across ages for both groups despite the expectation that children with unilateral hearing loss may have better FLI-P scores as they have one ‘good ear’ and results would be more similar to typical hearing children. $R^{2}$ values indicate similarly strong linear relationships, with a mildly stronger relationship for children with unilateral hearing loss ($R^{2}$ = 0.79) than for children with bilateral hearing loss ($R^{2}$ = 0.69).

The range of listening skills for children with unilateral hearing loss reflects the mixed outcomes for children with unilateral hearing loss reported in the literature. Reviews indicate speech and language delays in some but not all studies [16,17], difficulties at school with 22 to 35% of children with UHL repeating at least one grade, 12 to 41% receiving additional educational assistance [18], and poor levels of auditory performance [19,20].

#### 3.3.3. Additional Needs

The FLI-P scores for children in the EI group with additional disabilities (*n* = 92, 529 data points) were compared to children with hearing loss alone, regardless of the level of hearing loss or device (*n* = 315, 1706 FLI-P scores) (Figure 8). Results for the group with additional disabilities indicated the expected difference in listening skills, below the TH group, and below that of the hearing loss alone group. FLI-P scores for children with a disability in addition to hearing loss indicated greater variation ($R^{2}$= 0.59) than for the hearing loss alone group ($R^{2}$ = 0.70), in line with the evidence of the impact of an additional disability on the outcomes of children with hearing loss [21,22].

#### 3.3.4. Auditory Neuropathy Spectrum Disorder

The FLI-P scores for children with bilateral ANSD (*n* = 13, 127 FLI-P scores) were compared to children with unilateral ANSD (*n* = 20, 112 FLI-P scores). Children in each of these groups used a range of devices due to the individual nature of each child’s neuropathy. For children with bilateral ANSD this included two hearing aids (*n* = 1), one cochlear implant and one hearing aid (*n* = 1), and bilateral cochlear implants (*n* = 10). The 20 children with unilateral ANSD had no hearing loss in the other ear. These children wore cochlear implants (*n* = 3), bone conductor hearing aids (*n* = 3), a conventional hearing aid (*n* = 1), unaided (*n* = 1), and not recorded (*n* = 12).

Results indicate similar listening skills by ages for children with both bilateral and unilateral ANSD (Figure 9). A further analysis excluding children with an additional disability indicated less variation, as would be expected. Comparisons by age of implant indicated lower functional listening scores for children who received their first cochlear implant between 12 and 23 months than children who received their first implant between 6 and 11 months of age. Results indicated that the most closely matched FLI-P scores of the TH group were children with bilateral ANSD who received the earliest implants. The FLI-P results for children with ANSD in this study reflect the evidence of wide variability in outcomes [23,24,25]. Additionally, consistent with the literature, results demonstrated that a number of children with ANSD were developing listening skills aided with conventional hearing aids [26].

#### 3.3.5. Age at Diagnosis

The FLI-P scores for the children with hearing loss who were referred for diagnostic testing of hearing following newborn screening (*n* = 427, 2442 FLI-P scores) were compared to the FLI-P scores of children who passed newborn screening and were later diagnosed with a hearing loss (*n* = 54, 168 FLI-P scores). Information was not recorded in the database for 43 children (183 scores), and 19 children did not have their hearing screened at birth (76 FLI-P scores). For comparative purposes, Figure 10 displays the results of children with a bilateral moderate hearing loss or greater, referred through newborn hearing screening and had their first device fitted < 6 months of age, compared with children who passed newborn hearing screening and had their first devices fitted > 12 months of age. Greater variability in FLI-P scores is evident across all age ranges for children who passed newborn hearing screening. Given the amount of time children who are later diagnosed may have had without aiding and necessary access to sound, lower listening scores and higher levels of variability in scores would be expected.

The FLI-P scores of children diagnosed with a hearing loss following newborn screening in this study is consistent with the large body of evidence that early diagnosis enables the early development of auditory skills to support language acquisition [27,28,29]. In this study, variation in listening skill development as measured by the FLI-P was observed particularly for children at older ages who passed screening. Given the potential time gaps between the onset of the hearing loss following screening and diagnosis/fitting of devices, these FLI-P scores match the expected pattern.

#### 3.3.6. Type of Device

FLI-P scores were compared by type of hearing device. Devices were categorized according to bilateral cochlear implants (*n* = 96, 628 FLI-P scores), cochlear implant and hearing aid (*n* = 21, 130 FLI-P scores), bilateral hearing aids (*n* = 177, 864 FLI-P scores), and bilateral bone conductors (*n* = 6, 43 FLI-P scores). FLI-P scores for nine children (25 FLI-P scores) were excluded as devices were unknown. Results indicated no clear patterns between listening scores and devices (Figure 11). These data suggest that the device type a child uses is unlikely to be associated with listening outcomes, but instead their access to sound through whichever device is critical.

#### 3.3.7. Level of Hearing Loss

FLI-P scores were compared by the level of hearing loss across age groups. Hearing loss was grouped in five categories: mild–moderate and mild–profound (bilateral *n* = 114, 551 FLI-P scores), moderate and moderate–severe (bilateral *n* = 59, 381 FLI-P scores), severe and severe–profound (bilateral *n* = 20, 74 FLI-P scores), and profound (bilateral *n* = 44, 323 FLI-P scores). Children with high frequency hearing loss only were grouped in the mild and mild to profound group for the purposes of the analysis, and children with asymmetrical hearing losses were excluded due to the difficulty categorizing their hearing loss into a comparable group. FLI-P results for children with no level of hearing loss recorded in the database were excluded for 36 children (left ear) and 38 children (right ear) (Figure 12).

It has been consistently reported that the outcomes for children with hearing loss are impacted by the level of hearing loss [30,31]. These results in this study do not support such findings. Although the widest variability can be seen for children with profound hearing loss, levels of variability were observed across all levels. Consistent with the literature that early cochlear implants can result in age-appropriate outcomes, some FLI-P scores in both the severe and profound groups were commensurate with those in the TH group [32,33,34]. Lower FLI-P scores may well have been associated with other known factors to outcomes [35,36]. Further analysis of these groups accounting for known factors would be useful in understanding the full cause of the variability. FLI-P scores of children with different hearing levels in this study suggest that the level of hearing loss may not be as strong an impacting factor as, for example, age of access to appropriate levels of sound.

#### 3.3.8. Age at Implant

FLI-P scores were analyzed by age of implantation, a recognized factor impacting outcomes [37,38]. Listening scores for children with bilateral profound hearing loss using cochlear implants were grouped according to age of first implant: over 24 months (*n* = 48, 238 FLI-P scores), 12–24 months (*n* = 25, 166 FLI-P scores), 6–11 months (*n* = 29, 200 FLI-P scores), and under 6 months (*n* = 16, 150 FLI-P scores) (Figure 13). FLI-P scores indicate that children receiving the youngest cochlear implants (under 6 months of age, $R^{2}$ = 0.85 and 6–11 months of age, $R^{2}$ = 0.80) demonstrate FLI-P scores most similar to the TH group ($R^{2}$ = 0.80) and consistent with the reported literature. The similarity of scores to the TH group appears to reduce with older implant ages, and as variability in listening scores increase. Linear relationships match variability levels: over 24 months ($R^{2}$ = 0.30), 12–24 months ($R^{2}$ = 0.56), 6–11 months ($R^{2}$ = 0.80), and <6 months ($R^{2}$ = 0.85).

## 4. Discussion

The aim of this study was to determine the feasibility of the FLI-P as a clinical measure for all children with hearing loss enrolled in an early intervention and cochlear implant program. To establish this, the research objectives explored were:Whether the FLI-P could be used successfully with the entire population of children with hearing loss in a clinical/educational setting;If children’s individual FLI-P trajectories change with time as would be expected;If the data for known groups demonstrated the expected differences.

Results on all three objectives indicate good preliminary support for the ongoing use of the FLI-P as a clinical measure in an early childhood service for children with hearing loss. Data demonstrated that the FLI-P can be used successfully with the entire population of children with hearing loss enrolled in clinical/educational setting. That is, it can be used with a full range of children including those with all levels and types of hearing loss, who use a range of devices, from diverse language backgrounds, and with developmental needs in addition to hearing loss. FLI-P scores for different groups identified the expected differences, i.e., between children with typical hearing and hearing loss, disabilities in addition to hearing loss, and by age of implant. Although no clear differences in FLI-P scores across ages were indicated for children with bilateral and unilateral hearing loss, by type of hearing loss, or device used, these findings support the importance of early access through appropriate amplification device/s.

### 4.1. Use of the FLI-P in a Clinical/Education Setting

During the 5-year period in which the study was conducted, the FLI-P was able to be used on all children with hearing loss who entered the early intervention and implant program, regardless of age, language, hearing loss, device, or presence of additional needs. This broad demographic demonstrates the potential universality of the FLI-P for all children from birth through to 6 years of age. Highly significant correlations were found between age and FLI-P score for both the TH and EI group, indicating the expected growth in listening skills over time. This is the first measure clinicians have been able to use that provides this general application across the population and over the critical first 6 years of development for each child. 

### 4.2. Changes to Children’s Individual Scores over Time

FLI-P scores were shown to be reflective of a both a child’s longitudinal development of listening skills as well as at any given point in time. When FLI-P scores over time were graphically analyzed, the resulting developmental ‘listening paths’ provided a listening trajectory for each child. Although this was not a specific focus on this study, there were consistent reports from parents and clinicians in the appreciation of the utility of the visual way in which the data could be seen and used to inform discussions and decisions. A child’s trajectory, when set alongside those of others with selected characteristics can, for example, provide an early indication of slower than expected progress for an individual context. This enables parents and professionals to consider possible contributing factors, including a child’s use and integration of sound, levels of input, language exposure, and potential changes in access. Early identification of an impacting factor to outcomes can result in earlier changes to intervention in order to support positive changes. 

### 4.3. Differences in Group Scores

The typical hearing (TH) group provided a benchmark for comparison of FLI-P scores with the EI group. FLI-P scores that most closely matched the TH group were children with the earliest access to sound, regardless of the level of hearing loss or type of device. As expected, the group of children with a disability in addition to hearing loss and the group with ANSD in one or both ears had the widest range of variability in FLI-P scores of any of the EI participant groups. Children’s individual FLI-P scores and group data recorded during routine clinical practice supported findings that the FLI-P is responsive to the expected differences between groups, such as children with a disability in addition to hearing loss, and age at implant. Data on the validation of the FLI-P support these findings of the differences between groups and reports on the internal validity of the FLI-P through its concurrent validity with similar listening assessments providing a comparison of use, and predictive validity of a child’s language skills as well as construct validity by differences in known groups (a manuscript describing this study is in preparation).

With respect to the level of hearing loss, the FLI-P scores of the EI group with lower levels of hearing loss (unilateral, mild and moderate) still indicated a considerable range in scores when compared to those with greater levels of hearing loss and the most variation. Although these findings are not well documented elsewhere, they do reflect clinical experience that children with less significant levels of hearing loss (mild and moderate) do not necessarily experience the consistent, early, optimal access to sound as children with, for example, profound hearing loss who have audiological and educational intervention from a very early age. Future comparisons of FLI-P data with larger controlled cohorts across hearing levels could further explore this finding. 

### 4.4. Clinical Implications and Limitations

Comparative analyses in this study concentrated primarily on audiological and demographic characteristics. Analysis of group data beyond the scope of this work would provide additional information. Factors such as hearing loss etiologies, molecular diagnosis, specific gene mutations, gender, and cognitive and psychosocial profiles would be ideal areas for further investigation. Opportunities to better understand the impact of linguistic input and language learning environments on children’s developing listening skills, through the use of data logging and language environment analysis technologies, could be of substantial benefit in considering how to optimize a child’s language learning context. The listening development of children in different multilingual settings could also be explored.

Greater analysis of the listening skills of children with unilateral hearing loss and their corresponding language skills would also provide interesting insights to guide intervention practices. For example, a child who demonstrates scores significantly below the predicted trajectory or age norms for their particular characteristics (despite average language skills) may generate a review of their access to sound. Current practice generally defines that language scores for a child with hearing loss within the typical range indicate adequate access and adequate progress. Their potential though, may be much more. A sensitive measure of listening skills over time, such as the FLI-P, could contribute to supporting all children in being optimally amplified and reaching their language and communication potential—which may well be more than average. This could also provide critical information for children with unilateral hearing loss about access to sound for families making amplification choices, given the lack of current evidence and the challenges in relying on standardized language assessments to provide such guidance for intervention. Normative data are needed to further enhance the validity and use of the FLI-P and provide benchmark data for the listening skills of typical hearing children using the FLI-P to support improved clinical case management. A normative study has been completed on 451 children with typical hearing in the age range 1 to 72 months. A manuscript presenting results of this study is in preparation.

Undertaking this study across the population of children in an early intervention setting enabled access to valuable whole-of-clinic data rather than recruited, self-selected participants that may bias the data in unknown ways. The generalizability of results given the wide use across the entire clinical population is high, and the inclusion of all data in the study and lack of data exclusion minimized the risk of experimenter bias. As is common in the record keeping of a clinical service, missing data may have added elements of an inherent bias, and as the results were used clinically and in real-life practice, interventions were adjusted to address poor FLI-P scores, resulting in dynamically changing trajectories over time. Reports by clinicians of the sensitivity of the FLI-P to pick up impacting factors to develop (for example drops in hearing, middle ear infections) and use thereby to guide intervention is worth further analysis. Standards for use of the FLI-P need to be defined to increase reliability across users (for example the introduction of basal and ceiling limits and definitions for acquisition rating of skills with the yes/no categories on the FLI-P). Multicenter studies would indicate the generalizability of the FLI-P over different clinical and educational settings. While some data were collected from typical hearing participants, this number was small. 

## 5. Conclusions

Understanding how a child with hearing loss detects, uses, and processes linguistic input in their everyday settings, that is, their functional listening skill, is critical to understanding how well they are able to develop oral language. This study reports the results of 5 years of use of the Functional Listening Index (FLI-P) in a clinical retrospective study to identify its feasibility and viability as a measure of functional listening at individual points and over time. Results established that the FLI-P is usable across the population of children in a clinical/education setting regardless of individual demographic or audiological characteristics. Individual children’s FLI-P scores demonstrated the expected changes over time and showed sensitivity to factors known to impact listening development. Expected differences were seen in comparisons of group data and a dataset of typical hearing children provided an initial benchmark of expected listening development for comparison. Information provided by children’s listening scores on the FLI-P can guide and support discussions and intervention decisions and bridge the gap between information provided by audiological assessments of hearing levels and language measures. 

## Figures and Tables

**Figure 1 jcm-11-02764-f001:**
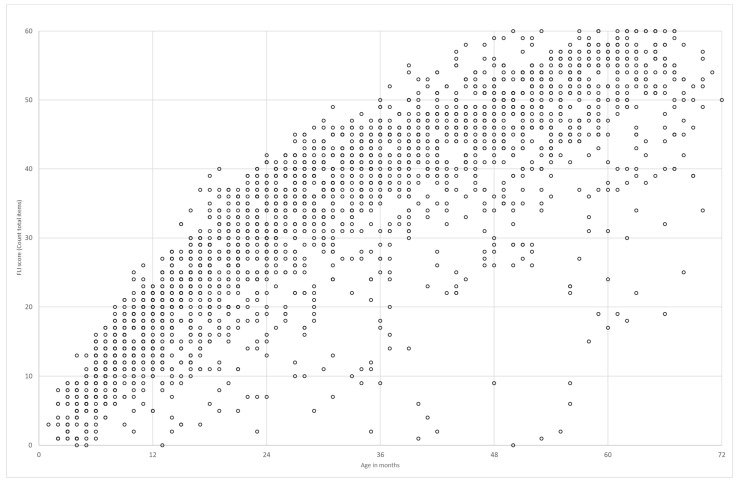
Early Intervention (EI) group total FLI-P data points. FLI-P scores collected from all children enrolled in the program over the course of the study. Each point may represent more than 1 child (if two children have identical FLI-P scores) and each child may have a number of scores (The original FLI, on which the FLI-P is based, contained 60 items. After initial reviews, an additional 4 items were added to the FLI-P as shown in Appendix A).

**Figure 2 jcm-11-02764-f002:**
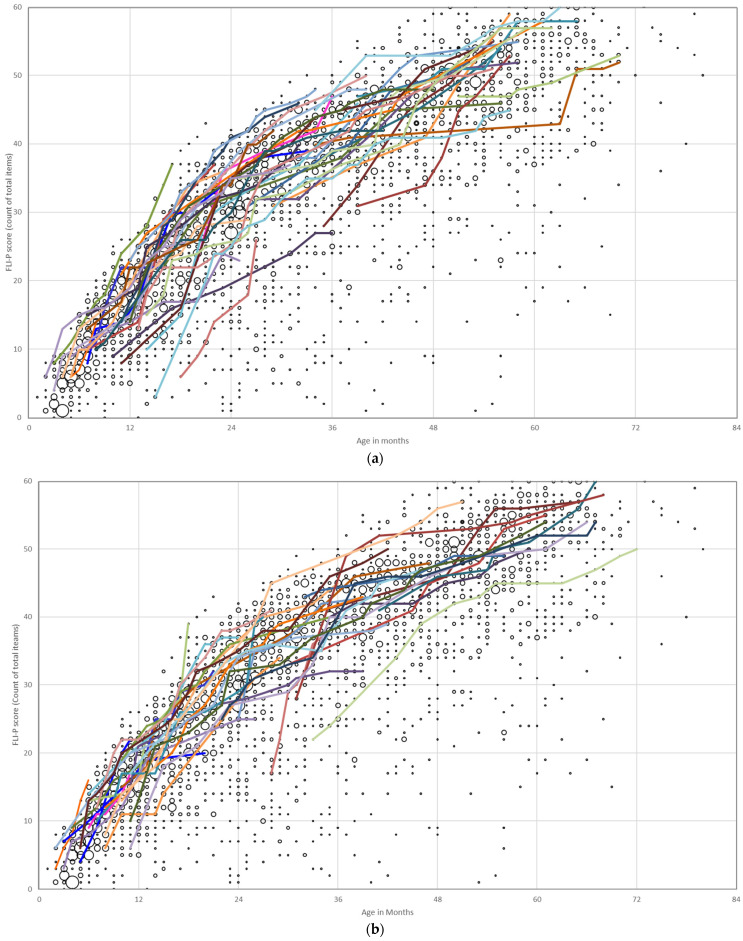
Individual listening trajectories of FLI-P scores of children with bilateral (**a**) mild, mild–moderate, and mild–profound (41 children); (**b**) moderate and moderate–severe (33 children); (**c**) severe and severe–profound (16 children); and (**d**) profound hearing loss (30 children). Each of the colored lines represents an individual child’s trajectory to highlight progress and different listening development trajectories.

**Figure 3 jcm-11-02764-f003:**
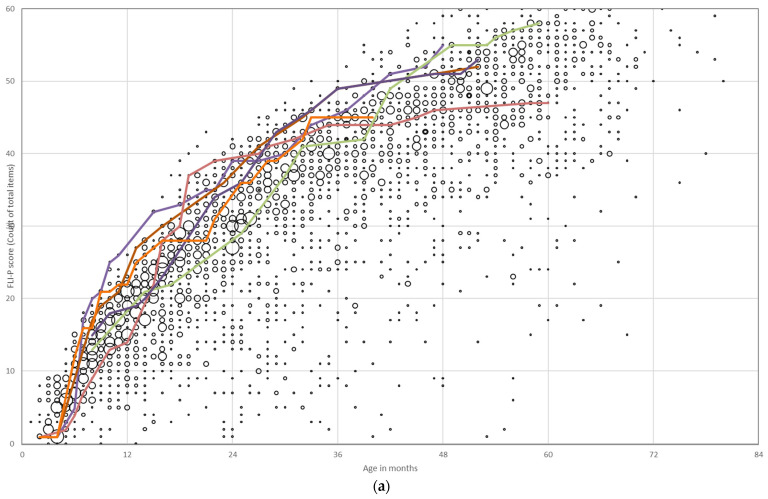
Individual listening trajectories of FLI-P scores of children with bilateral profound hearing loss who received their first cochlear implant (**a**) under 6 months of age; (**b**) between 6 and 11 months of age; and (**c**) between 12 and 23 months of age. Scores show a steep increase in skills post implantation and ongoing acquisition with age. Each of the colored lines represents an individual child’s trajectory to highlight progress and different listening development trajectories.

**Figure 4 jcm-11-02764-f004:**
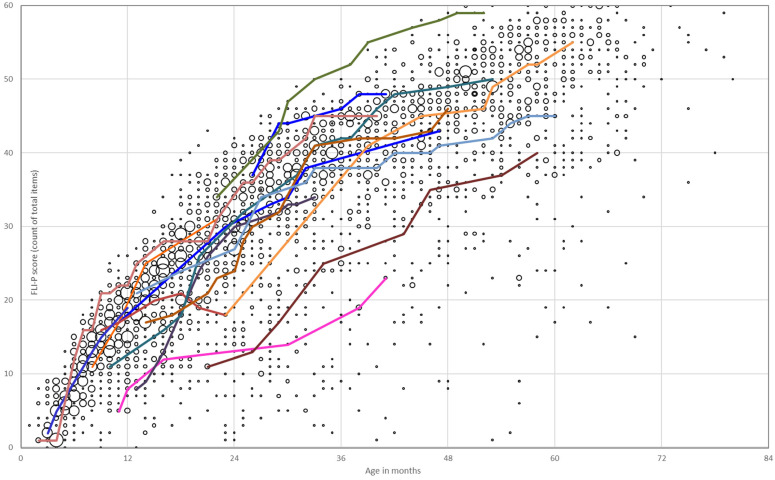
Individual trajectories of FLI-P scores for children with cochlear implants, bilateral severe, severe–profound, and profound hearing loss and a diagnosed additional disability. Each of the colored lines represents an individual child’s trajectory to highlight progress and different listening development trajectories.

**Figure 5 jcm-11-02764-f005:**
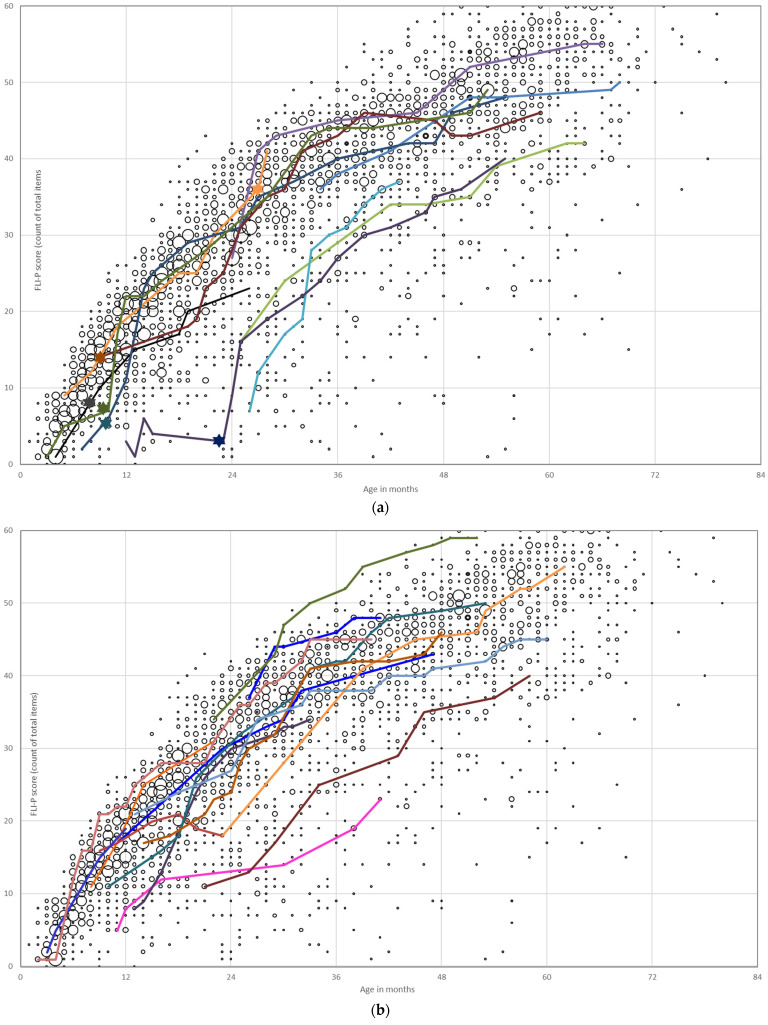
Individual trajectories of FLI-P scores for children with (**a**) bilateral ANSD and (**b**) unilateral ANSD. Results show a range in listening skill development as expected with ANSD. Increases in listening skills are evident post cochlear implantation (large dots). Each of the colored lines represents an individual child’s trajectory to highlight progress and different listening development trajectories.

**Figure 6 jcm-11-02764-f006:**
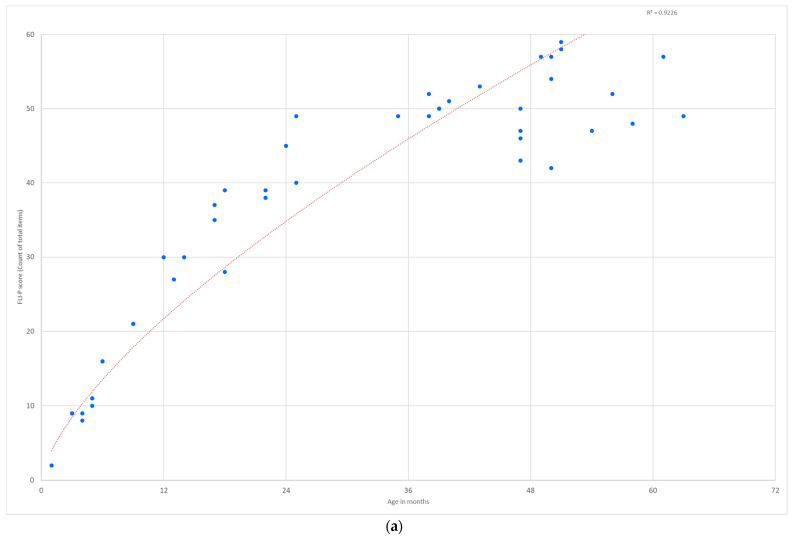
FLI-P scores (**a**) for children in the typical hearing group (*n* = 27 children, 46 data points) show an increase in listening skills with age and variability in scores for children at 4 years of age and (**b**) comparative scores for the early intervention group (*n* = 451 children, 2340 data points).

**Figure 7 jcm-11-02764-f007:**
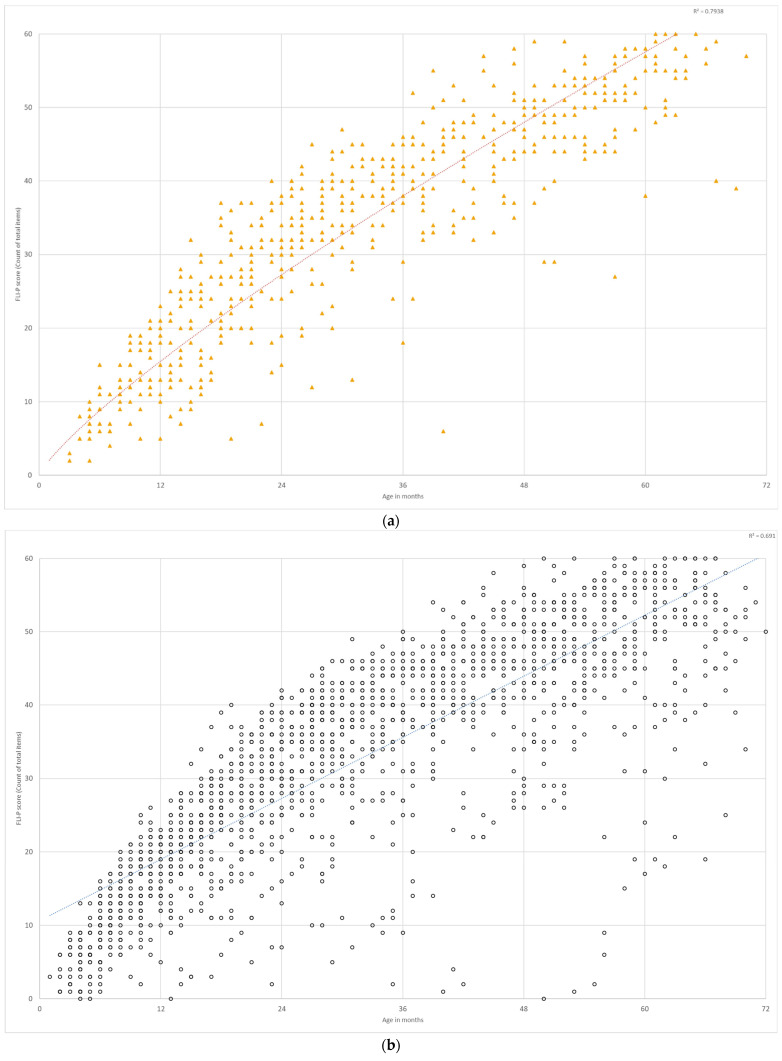
FLI-P scores of children with (**a**) unilateral HL and (**b**) bilateral HL.

**Figure 8 jcm-11-02764-f008:**
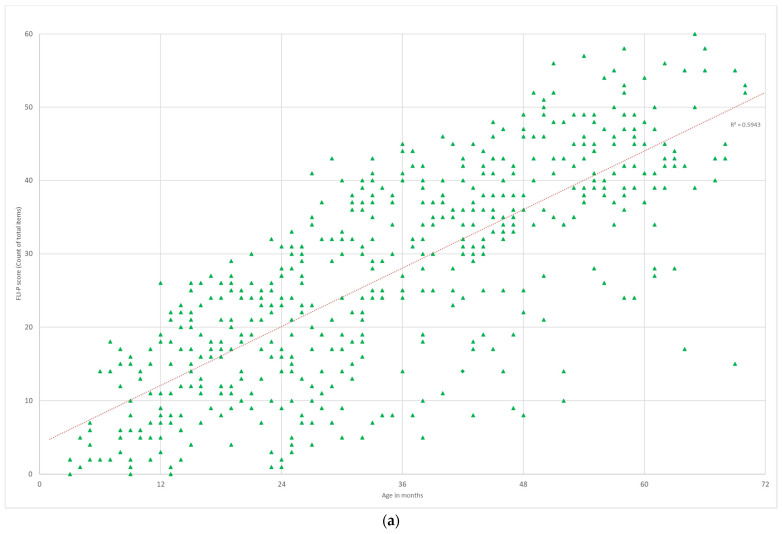
FLI-P scores between children with (**a**) the presence of an additional need to hearing loss and (**b**) no additional need.

**Figure 9 jcm-11-02764-f009:**
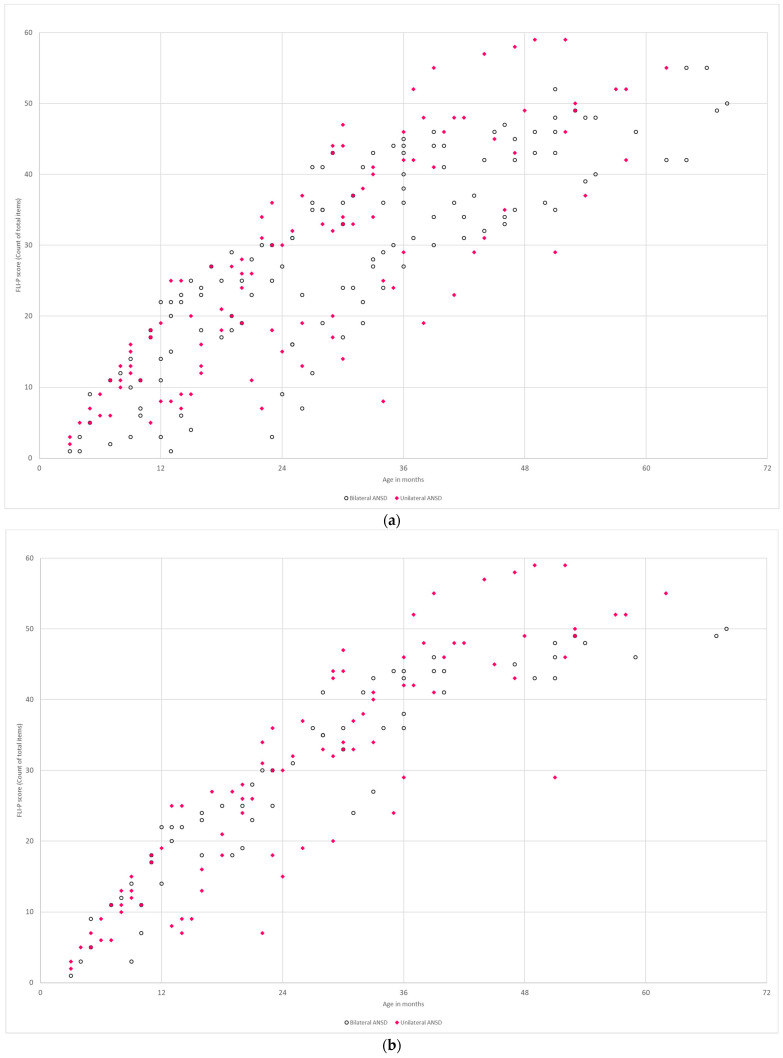
FLI-P scores (**a**) bilateral ANSD (no color) vs. unilateral ANSD (colored dots); (**b**) bilateral ANSD (no color) vs. unilateral ANSD (colored dots), no additional development needs; and (**c**) bilateral ANSD with a CI between 6 and 11 months (*n* = 5) (no color) and 12 and 23 months (*n* = 6) (colored dots).

**Figure 10 jcm-11-02764-f010:**
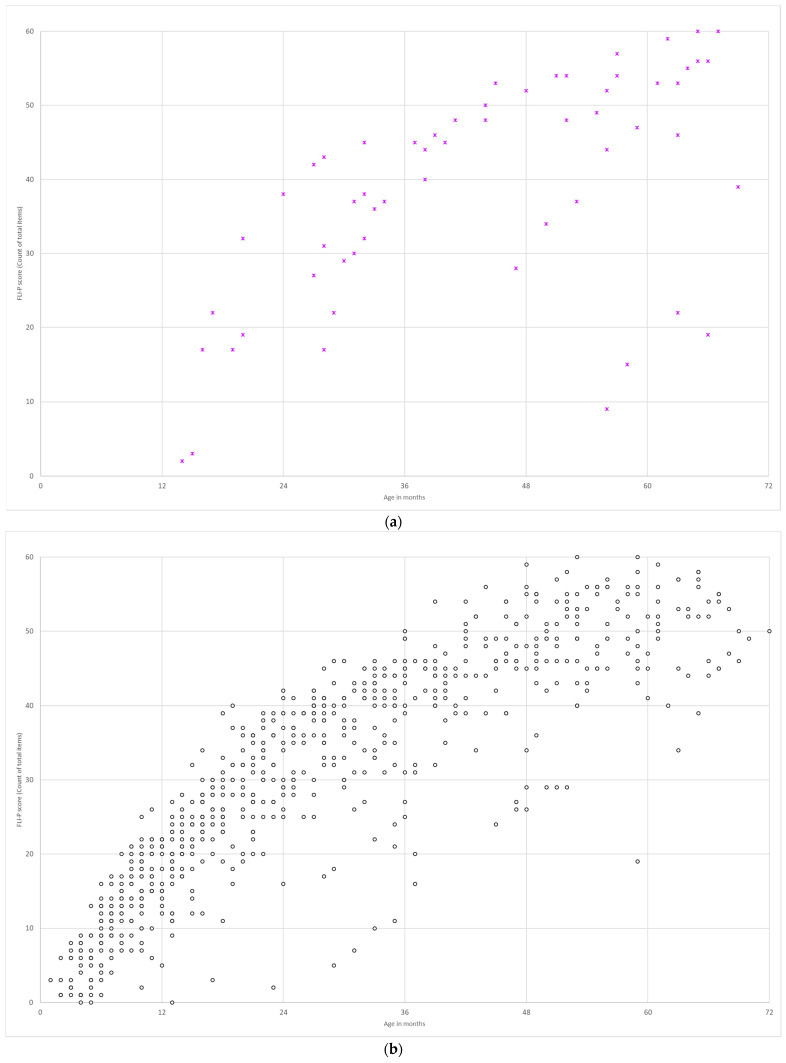
FLI-P scores between groups of children who (**a**) received a refer newborn hearing screening result and (**b**) who received a result.

**Figure 11 jcm-11-02764-f011:**
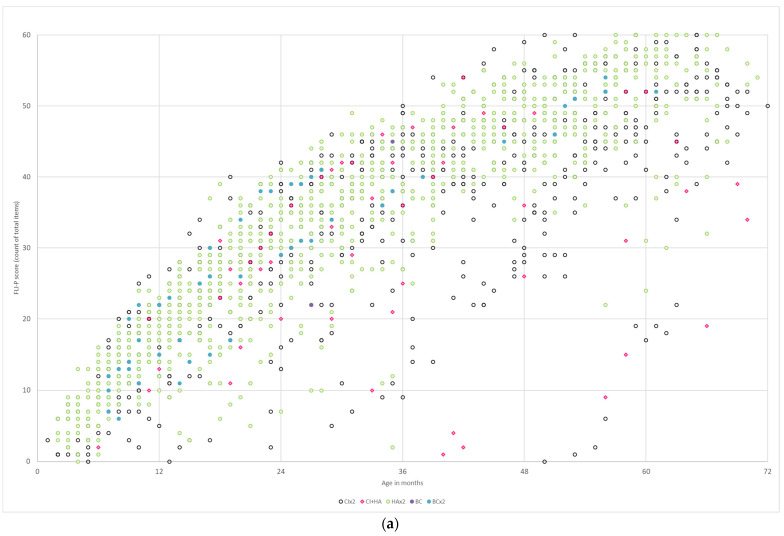
FLI-P scores by device for children with (**a**) bilateral hearing loss with bilateral cochlear implants (no color), a cochlear implant and a hearing aid (red), bilateral hearing aids (green), a bone conduction device (black), and bilateral bone conduction devices (blue); and (**b**) unilateral hearing loss with a cochlear implant (blue), a hearing aid (no color), and a bone conduction device (yellow). There were no observable patterns in either group.

**Figure 12 jcm-11-02764-f012:**
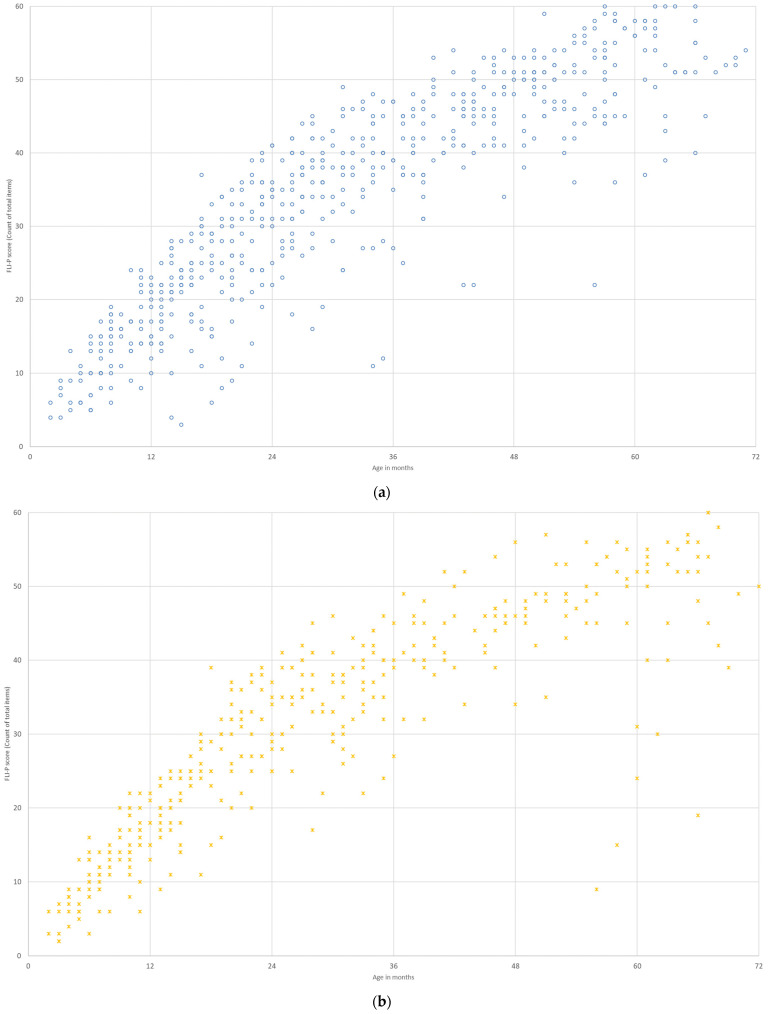
FLI-P scores by (**a**) mild and mild–profound; (**b**) moderate and moderate–severe; (**c**) severe and severe–profound; and (**d**) profound bilateral (no additional needs).

**Figure 13 jcm-11-02764-f013:**
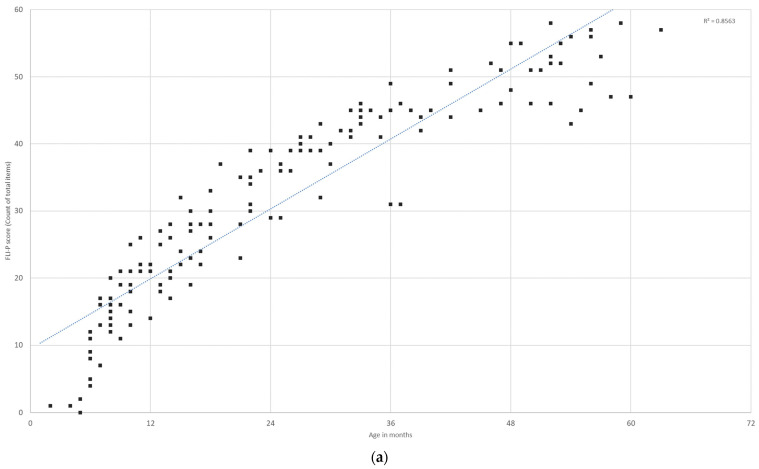
FLI-P scores for children with bilateral profound hearing loss by age at implant (**a**) <6 months; (**b**) 6–11 months; (**c**) 12–23 months; and (**d**) >24 months.

**Table 1 jcm-11-02764-t001:** Early Intervention (EI) group participant characteristics.

EI Group	Characteristic	Number of Children
Gender	Male	285
	Female	258
Symmetry of hearing loss (HL)	Bilateral	385
	Unilateral	140
	Unknown	18
Presence of additional needs that impact on learning	Yes	92
	No	451
Newborn Hearing Screening	Pass	54
	Refer	427
	Not tested	19
	Unknown	43
Age at diagnosis (months)	<3	380
	3–6	23
	7–11	15
	12–23	37
	24–36	23
	>36	33
	Unknown	32
Age at first device fitting (months)	<3	177
	3–6	73
	7–11	34
	12–23	81
	24–36	44
	>36	51
	Unknown	83
Age at entry to EI (months)	<3	168
	3–6	96
	7–11	70
	12–23	86
	24–36	53
	>36	70
	Unknown	0

**Table 2 jcm-11-02764-t002:** Early Intervention (EI) group (FLI-P data points) by type of bilateral hearing loss.

		Left				
		Conductive	Sensorineural	Mixed	Unknown	* ANSD
Right	Conductive	23 (88)	3 (23)	1 (2)	n/a	n/a
	Sensorineural	1 (3)	286 (1616)	2 (6)	1 (1)	2 (5)
	Mixed	2 (20)	3 (6)	24 (138)	1 (1)	n/a
	Unknown	n/a	n/a	n/a	15 (45)	n/a
	ANSD	n/a	1 (6)	n/a	n/a	13 (127)
	Normal	n/a	4 (19)	n/a	n/a	n/a

* Auditory Neuropathy Spectrum Disorder.

**Table 3 jcm-11-02764-t003:** Early Intervention (EI) group (FLI-P data points) by type of unilateral hearing loss.

EI Group	Type	Number of Children
Left	Conductive	14
	Sensorineural	37
	Mixed	5
	Unknown	1
	Auditory Neuropathy Spectrum Disorder	11
Right	Conductive	26
	Sensorineural	29
	Mixed	7
	Unknown	1
	Auditory Neuropathy Spectrum Disorder	9

**Table 4 jcm-11-02764-t004:** Early Intervention (EI) group by device used (bilateral hearing loss).

EI Group	Type	Number of Children
Left	Cochlear implant	138
	Hearing aid	218
	Bone conduction device	13
	Unknown	14
	Unaided	1
Right	Cochlear implant	146
	Hearing aid	215
	Bone conduction device	9
	Unknown	15
	Unaided	0

**Table 5 jcm-11-02764-t005:** Early Intervention (EI) group by device used (unilateral hearing loss).

EI Group	Type	Number of Children
Left	Cochlear implant	11
	Hearing aid	15
	Bone conduction device	15
	Unknown	30
Right	Cochlear implant	10
	Hearing aid	18
	Bone conduction device	21
	Unknown	31

**Table 6 jcm-11-02764-t006:** Early Intervention (EI) group by level of bilateral hearing loss *.

EI Group	Type	Number of Children
Left	Normal	81
	High frequency	8
	Mild	52
	Mild–moderate to profound	106
	Mod–moderate/severe to profound	122
	Severe, severe–profound	58
	Profound	80
	Unknown	36
Right	Normal	70
	High frequency	8
	Mild	50
	Mild–moderate to profound	115
	Mod–moderate/severe to profound	128
	Severe, severe–profound	49
	Profound	85
	Unknown	38

* Standard audiometric terminology has been used (categorized according to hearing in the better ear and defined as mild = 26–40 dB, mild–moderate to profound = 26–91 dB, mod–moderate/severe to profound = 56–91 dB, severe, severe–profound = 71–91 dB, and profound ≥ 91 dB) across 500 Hz, 1000 Hz, 2000 Hz, and 4000 Hz. High frequency losses include those in the mild through to profound categories.

**Table 7 jcm-11-02764-t007:** Early Intervention (EI) group by level of unilateral hearing loss *.

EI Group	Type	Number of Children
	Normal	70
	High frequency	1
	Mild	6
	Mild–moderate to profound	6
	Mod–moderate/severe to profound	25
	Severe, severe–profound	19
	Profound	9
	Unknown	4

* Standard audiometric terminology has been used (categorized according to hearing in the better ear and defined as mild = 26–40 dB, mild–moderate to profound = 26–91 dB, mod–moderate/severe to profound = 56–91 dB, severe, severe–profound = 71–91 dB, and profound ≥ 91 dB) across 500 Hz, 1000 Hz, 2000 Hz, and 4000 Hz. High frequency losses include those in the mild through to profound categories.

## Data Availability

Data supporting reported results can be found by contacting the lead author.

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
