# Peer review of "The Feasibility of the Functional Listening Index—Paediatric (FLI-P®) for Young Children with Hearing Loss"

_jcm, 2022, doi:10.3390/jcm11102764_

Round 1

Reviewer 1 Report

Overall, the manuscript was well written, the methods were sound, and the result presentation was clear. I only have a few minor points.

Line 21-23 Repetitive

Line 231. Define large and small dots in the figure caption.

Table 5 & Table 6. I found that terms like mild-moderate, mild-profound, moderate-profound, moderate-severe are confusing. It may be better to use mild I, mild II, etc. instead. At least they need to be defined in section 2.6.

Section 3.3.1. Why using linear fit? In both TH and EI groups it seems that there is a curvilinear trend for the FLI-P score as a function of age.

Figure 3. What are the colored lines?

Figure 9. Define color dots and grey dots in the caption.

Figure 11. Define the color scheme for the four device groups in the caption.

Author Response

Response to Reviewer 1’s comments

The authors greatly appreciate the editorial input from the reviewers in moving this manuscript to publication. As per journal instructions, changes have been tracked in the link provided, and responses to each point detailed below.

Point 1: Minor spell check required.

Response 1: We thank the reviewer for the comment. We have rechecked the spelling of the complete document to ensure consistency. For clarity British English has been used throughout the paper (for example colour and paediatric) and the Oxford Style Guide for -ize endings (for example analyzed, standardized, individualized).

Point 2: Results presentation can be improved.

Response 2: We thank the reviewer for their comments regarding the presentation of the results. We have made changes as outlined in the below points, which we believe increases the clarity  regarding the presentation and labelling.

Point 3: Line 21-23 repetitive

Response 3: The repeated words (different ages, degrees of hearing loss, device use, and additional needs) have been deleted in lines 24 – 25.

Point 4: Line 231: Define large and small dots in the figure caption

Response 4: We thank the reviewer for picking up the need for clarity with these dots in Figure 2 c) and d). This has been clarified both in the text and in the description under the Figures. There is now an explanation that the large dots in Fig 2c) represent points of cochlear implantation. The small dots in the other graph have been made removed for consistency with Figures 2 a) and b).

Point 5: Tables 5 & 6: Category labels are confusing. Suggest Mild 1, mild II etc. Define in Section 2.6

Response 5: Thank you for this comment. The categories have been clarified in the note under each relevant Table (Tables 6 and 7) lines 170 – 173 and 175 – 178 to state that standard audiometric terminology has been used and definitions provided for the level of loss in each category. In looking at this we also discovered Table 5 was originally mislabelled so this has also been corrected.

Point 6: Section 3.3.1. Why using linear fit? In both TH and EI groups it seems that there is a curvilinear trend for the FLI-P score as a function of age.

Response 6: This is an excellent point from the reviewer. We have reviewed the graphs and applied a curvilinear fit to all relevant graphs which now demonstrates periods of rapid acquisition of skills and a more significant  value (Figs 6 a), 7 a)) and added  values on Fig 8 a) and b) for consistency.

Point 7: Fig 3: What are the colored lines?

Response 7: Thank you for highlighting the need to clarify the coloured lines. Text has now been added under each figure with coloured lines, and in the corresponding text where appropriate, to explain that ‘each of the coloured lines represents an individual child’s trajectory to highlight progress and different listening development trajectories’ Lines 245 – 246, 256 – 258, 266 -267, 274 – 275, 287 – 289, and 301 – 303.

Point 8: Fig 9: Define color dots and grey dots in the caption

Response 8: As per the above comment, the coloured dots have been defined in the note below the figure (lines 382 – 385).

Point 9: Fig 11: Define the color scheme for the four device groups in the caption

Response 9: We thank the reviewer for this comment. The colour scheme for the four groups have been clarified in the caption for both Figures 11 a) and b) (Lines 421 – 426).

Reviewer 2 Report

Aleisha Davis and colleagues submitted a research article entitled: The Feasibility of The Functional Listening Index – Paediatric 2 (FLI-P®) for Young Children with Hearing Loss

They establish the feasibility of a tool to monitor the development of functional listening skills to inform early and ongoing decisions by parents and professionals.

The data are very interesting and could be of interest to the readers.

Minor comments:

  1. Was any molecular diagnosis performed for the patients?
  2. Can we link molecular diagnosis or mutation in a specific gene with the ability of the patient to respond to the given tests?
  3. Male and female comparative response to the study? Which were more active?
  4. Kindly list the limitations of the study.
  5. Study approval?
  6. The consent form was obtained from the participants?
  7. Did the author compare the Functional Listening Index with some other listing platform? any comparison?

Author Response

Response to Reviewer 2’s comments

The authors greatly appreciate the editorial input from the reviewers in moving this manuscript to publication. As per journal instructions, changes have been tracked in the link provided, and responses to each point detailed below.

Point 1: Moderate English changes required.

Response 1: We thank the reviewer for the comment. We have rechecked the spelling of the complete document to ensure consistency. For clarity British English has been used throughout the paper (for example colour and paediatric) and the Oxford Style Guide for -ize endings (for example analyzed, standardized, individualized).

Point 2: Research design can be improved.

Response 2: We’re thankful for this comment and agree that in a designed research study where participants are specifically selected, information collected and assessments/measures taken specifically for the purpose of research this would be the case.  However, as the purpose of this study was to examine  the feasibility of use of the tool in a real-life setting, the research design was limited to a study of effectiveness.  Validation and comparative analysis of the tool is a critical component for evidence-based use of the tool, and this work was conducted in parallel to the study presented in this manuscript.  We have included these validation studies in a further manuscript currently in preparation for submission as it was judged be too much information for one publication.

Point 3: Conclusions could be more supported by the results

Response 3:  Thank you again for this comment. As per the above point, a study designed specifically in controlled conditions to establish efficacy through causal relationships could further support findings, and are in place to supplement the findings reported here. 

Point 4: Can we link molecular diagnosis or mutation in a specific gene with the ability of the patient to respond to the given tests?

Response 4: We thank the reviewer for this point. This may well be possible. As molecular diagnosis or specific gene mutations are not routinely recorded in early intervention settings this data was not available for this study, although as per the above point, a controlled study with this as an area of focus could investigate this further. This has been added to the study limitations (Lines 550 - 551).

Point 5: Male and female comparative response to the study? Which were more active?

Response 5: Thank you for this question. The subject population of males and females in the study was representative of the population on which the tool is used. Internal analysis of outcomes have shown no gender bias to listening outcomes and as such was not investigated specifically as part of this study, although could be in future studies. This has now been identified in the limitation section (Lines 550 – 551). 

Point 6: Kindly list the limitations of the study.

Response 6: These have been listed between Lines 548 – 592.

Point 7: Study approval?

Response 7: Thank you for this question. As per the journal submission instructions this has been listed under the Institutional Review Board Statement (Lines 615 – 617).

Point 8: The consent form was obtained from the participants?

Response 8: We thank the reviewer for the question. Yes, this has been stated under the Information Consent Statement (Lines 618 - 619).

Point 9: Did the author compare the Functional Listening Index with some other listing platform? any comparison?

Response 9: Thank you, yes this is an excellent question in understanding the need for and effectiveness of the tool. As per Lines 533 -534, concurrent validity through comparison on the tool with similar assessments has been measured and the data for which is in preparation for submission as part of the validation of the FLI-P. The tools that the FLI-P was compared to in this parallel publication are the The Parents' Evaluation of Aural/Oral performance of Children (PEACH+) (Ching & Hill, 2005), and the LittlEARS Auditory Questionnaire (LEAQ) (Tsiakpini et al., 2004).

Reviewer 3 Report

The manuscript describes evaluation of a 64-item checklist, the FLI-P, used to monitor the development of listening skills in children with hearing loss (HL).  Summary data for a large N of participants are reported, broken down by typical factors such as degree of HL and length of cochlear-implant use.  A general finding was that the survey can be used with children having a range of clinical profiles; scores increased with age across all groups and sub-groups.  This preliminary report emphasizes patterns across groups.  It does not provide statistical validation or amount to a full clinical trial of the FLI-P, but it does suggest that more detailed efforts toward those goals might be warranted.

Author Response

Response to Reviewer 3’s comments

The authors greatly appreciate the editorial input from the reviewers in moving this manuscript to publication. As per journal instructions, changes have been tracked in the link provided, and responses  detailed below.

Point 1: The manuscript describes evaluation of a 64-item checklist, the FLI-P, used to monitor the development of listening skills in children with hearing loss (HL).  Summary data for a large N of participants are reported, broken down by typical factors such as degree of HL and length of cochlear-implant use.  A general finding was that the survey can be used with children having a range of clinical profiles; scores increased with age across all groups and sub-groups.  This preliminary report emphasizes patterns across groups.  It does not provide statistical validation or amount to a full clinical trial of the FLI-P, but it does suggest that more detailed efforts toward those goals might be warranted.

Response 1: We thank the reviewer for their comments. We look forward to reporting on the statistical validation, normative benchmark data and clinical trials of the FLI-P in future publications.